# Plasma-Treated Air and Water—Assessment of Synergistic Antimicrobial Effects for Sanitation of Food Processing Surfaces and Environment

**DOI:** 10.3390/foods8020055

**Published:** 2019-02-02

**Authors:** Uta Schnabel, Oliver Handorf, Kateryna Yarova, Björn Zessin, Susann Zechlin, Diana Sydow, Elke Zellmer, Jörg Stachowiak, Mathias Andrasch, Harald Below, Jörg Ehlbeck

**Affiliations:** 1Plasma Bioengineering, Leibniz Institute for Plasma Science and Technology e.V., Felix-Hausdorff-Straße 2, 17491 Greifswald, Germany; oliver.handorf@inp-greifswald.de (O.H.); ky153778@uni-greifswald.de (K.Y.); bjoern.zessin@web.de (B.Z.); susi-zechlin@gmx.de (S.Z.); diana.sydow@yahoo.com (D.S.); joerg.stachowiak@inp-greifswald.de (J.S.); andrasch@inp-greifswald.de (M.A.); ehlbeck@inp-greifswald.de (J.E.); 2School of Food Science and Environmental Health, College of Sciences and Health, Technological University Dublin, Cathal Brugha Street, D01 HV58 Dublin, Ireland; 3Institute for Hygiene and Environmental Medicine, Faculty of Medicine, University of Greifswald, Walter-Rathenau-Straße 49A, 17475 Greifswald, Germany; zellmere@uni-greifswald.de (E.Z.); below@uni-greifswald.de (H.B.)

**Keywords:** antimicrobial, atmospheric pressure, decontamination, ion chromatography, microwave, nitrate, nitrite, non-thermal plasma

## Abstract

The synergistic antimicrobial effects of plasma-processed air (PPA) and plasma-treated water (PTW), which are indirectly generated by a microwave-induced non-atmospheric pressure plasma, were investigated with the aid of proliferation assays. For this purpose, microorganisms (*Listeria monocytogenes*, *Escherichia coli*, *Pectobacterium carotovorum*, sporulated *Bacillus atrophaeus*) were cultivated as monocultures on specimens with polymeric surface structures. Both the distinct and synergistic antimicrobial potential of PPA and PTW were governed by the plasma-on time (5–50 s) and the treatment time of the specimens with PPA/PTW (1–5 min). In single PTW treatment of the bacteria, an elevation of the reduction factor with increasing treatment time could be observed (e.g., reduction factor of 2.4 to 3.0 for *P. carotovorum*). In comparison, the combination of PTW and subsequent PPA treatment leads to synergistic effects that are clearly not induced by longer treatment times. These findings have been valid for all bacteria (*L. monocytogenes* > *P. carotovorum* = *E. coli*). Controversially, the effect is reversed for endospores of *B. atrophaeus*. With pure PPA treatment, a strong inactivation at 50 s plasma-on time is detectable, whereas single PTW treatment shows no effect even with increasing treatment parameters. The use of synergistic effects of PTW for cleaning and PPA for drying shows a clear alternative for currently used sanitation methods in production plants. Highlights: Non-thermal atmospheric pressure microwave plasma source used indirect in two different modes—gaseous and liquid; Measurement of short and long-living nitrite and nitrate in corrosive gas PPA (plasma-processed air) and complex liquid PTW (plasma-treated water); Application of PTW and PPA in single and combined use for biological decontamination of different microorganisms.

## 1. Introduction

Various prevention strategies against contaminations of food produces are relevant among the European Union. Consequently, a vast number of now well-established practices and concepts such as good agricultural practice, good manufacturing practice and hazard analysis (HACCP) were designed to minimize the risk of contaminations with pathogenic bacteria and food spoilers. However, the whole value chain of fresh and fresh-cut produce harbors numerous sources of contaminations such as raw material, processing tools, surfaces with food contact, or insufficiently trained employees. Once contaminated it is very likely that a produce carries the contamination along the whole production chain. Therefore, the in-company process hygiene is under special surveillance and new, innovative techniques recently gain attention [1]. Nowadays the situation is aggravated by the fact that inconsistent food safety standards were developed into economic, political, and social problems. The reason for that confusingly amount of sanitation standards is especially due to the presence of new microbial loads such as altered focuses on accurately described bacteria or bacteria that conquered new habitats, which might alter locally. However, the main body of sterilization and disinfection methods based on wet chemical processes (peracetic acid, hydrogen peroxide, chlorine dioxide, chlorine, active oxygen, ammonium compounds, isopropanol, phosphoric acid, formic acid or nitric acid), which are highly effective but also increase the development of new resistances [1]. Certainly, cleaning and disinfection cycles that clean the production environment in terms of the removal of organic (proteins, lipids, carbohydrates) and inorganic (salts) contaminations were integrated into every process chain. Nevertheless, it will be a future challenge to develop new maybe unified safety standards, which base on alternative techniques and may help to overcome those economic, political, and social barriers. The complex geometries of food processing plants made from various materials (stainless steel, glass, polymers, rubber and polytetrafluoroethylene (PTFE)) make effective cleaning difficult. For instance, empirically designed routine processes can lead to incomplete microbial decontamination, so that pathogenic microorganisms such as *Listeria monocytogenes* can survive on surfaces and contaminate the further process chain [2,3,4]. Under these circumstances, the risk of a microbial adhesion and biofilm formation is particularly high. Cross-contaminations, bacterial adhesion, and biofilm formation was avoided by an optimized choice of materials, which can be further achieved by efficient and sustainable agents with antimicrobial effects [5,6]. Manufacturers are therefore faced with a wide range of resources-consuming and cost-intensive challenges (environment, energy and water consumption, residues as well as storage and disposal etc.), which raised the demand for innovative process concepts that supplement or replace existing systems. When the food production is resumed after the heat decontamination step, the cooling time appears to be especially crucial. Certainly, active cooling is possible, but it is also energy consuming. 

At boundaries between two materials with a different thermal expansion, e.g., at observation windows, heat decontamination induces tensions that may lead to cracks in the glass window. For instance, windows for optical sensors like in-line process photometers for in-line phase switch control may be such a weak point. Particularly, calcifications will become critical, if heat decontamination is combined with water vapor [7]. When heat have been used for decontamination, the chamber where the decontamination takes place needs to be pre-heated. Due to the low heat transfer coefficient for the heat transfer between gaseous and solid phases and the high heat capacity of steel in contrast to ambient air, stainless steel part needs a long time to heat up or cool down. Therefore, colder plant parts lower the surrounding temperature, which may result in an insufficient decontamination rate, a pre-heating appears to be a very crucial step. Additionally, overpressure has the drawback to increase the mechanical load on machine parts in contact, which increases the frequencies of maintenance for those plants [7]. Therefore, all machine components have to be pressure-proofed. If heat is used these drawbacks lack when PPA (plasma-processed air) is applied. The use of PTW (plasma-treated water) enables the user to apply the shearing forces of the water for cleaning; which is not possible in decontamination steps using gaseous compounds. Additionally, when using heat, special attention must also be paid to the work safety of the user, i.e. to prevent burns. This is not necessary with the plasma technology presented here where the work safety requirements are easy to implement. Despite many advantages, plasma technology is an individual solution for special requirements and can currently be seen as a niche product.

Several studies prove the efficiency of non-thermal atmospheric pressure plasma applications for a surface decontamination. The resulting reactive species (including oxygen and hydroxyl radicals, nitrogen oxides and other oxidizing species) proved to be highly effective against many yeasts, viruses and bacteria (including spores). The results have been found on various surfaces such as glass, polypropylene (PP), paper or stainless steel [8,9]. The exact inactivation mechanism of the different plasma sources has already been the subject of several reviews [10,11]. The removal of biomolecules such as proteins and peptides, which are also required for purification, is also possible by plasma application [12].

For the indirect plasma process based on PLexc microwave plasma, antimicrobial effects have already been demonstrated on both laboratory and industrial scale on abiotic surfaces (glass, plastic) and biological surfaces (fruit, vegetables, meat) [13,14,15,16,17,18]. It applies to both PPA and PTW.

Currently, synergistic effects of both methods have not yet been deeply investigated and are the object of this study. Our findings strongly suggest a serial application of the two methods. If the production process allows such an application, a PTW rinsing/washing step should be followed by a PPA- drying step, subsequently. Many processes in the food industry fulfill these requirements, which give the PTW/PPA processes a wide range for their industrial application. 

## 2. Materials and Methods

### 2.1. Microorganisms

To cover a wide range of microbial germs, like bacteria, fungi and spores as well as human and plant pathogens occurring in connection with food and food production, *Pectobacterium carotovorum* (ATCC 15713, American Type Culture Collection), *Escherichia coli* K12 (NCTC 10538, The National Collection of Type Cultures), *Listeria moncytogenes* (ATCC 15313), and endospores of *Bacillus atrophaeus* (ATCC 9372) were used for microbial investigations. All bacteria and spores were obtained from Leibniz Institute DSMZ-German Collection of Microorganisms and Cell Cultures, Braunschweig, Germany. 

The bacterium *P. carotovorum* is a Gram-negative ubiquitous plant pathogen responsible for bacterial soft rot with a wide host range such as leafy greens, potato, carrot, tomato and many more. The bacterium *L. monocytogenes* is a Gram-positive human pathogen, which causes listeriosis infections under invasive conditions. Under non-invasive circumstances, any infection is termed febrile gastroenteritis. It is one of the most virulent foodborne pathogens, with high fatality (20 to 30%) in high-risk individuals including pregnant women, the elderly and individuals with compromised immune systems [19,20]. The bacterium *E. coli* is a Gram-negative mostly harmless part of the guteral normal intestinal flora of warm-blooded organisms like humans [21,22,23]. Therefore, it was expelled into the environment within fecal matter [24]. Some serotypes of *E. coli* have been known to cause serious food poising, which is occasionally responsible for food contamination, which leads to product recalls [25,26]. *E. coli* K12 is a common laboratory strain with low handling risks due to a mutation that prevents O-antigen formation. Endospores of *Bacillus atrophaeus* are test strains for EO (ethylene oxide) and H_2_O_2_ sterilization [27,28] as well as biological indicator for hot air sterilization. They are highly resistant to many disinfectants. Endospores are inactive or dormant forms of these bacterial strains and can resist highly disadvantageous conditions to reactivate to vegetative cells whenever the circumstances improve. 

Before use, the bacteria was stored at −80 °C and freshly cultivated in nutrient broth and on nutrient agar at specific incubation temperatures as required by them (Table 1). The spores were stored at 7 °C and used directly in dilutions with NaCl solution (0.85%). 

For experimental use, one inoculation loop of inoculated broth was cultured on agar for 24 h at specific incubation temperature and afterwards it was stored at 7 °C until use. The maximum storage time was 3 weeks.

### 2.2. Specimen

The microorganism was grown on biocompatible polyethylene (PE) polymeric stripes in the size of 32 × 8 × 2 mm^3^. PE is a polymer that is frequently used in food industry for conveyer belts, transport or collecting boxes, and packaging material. The specimen was contaminated by overnight cultivation of cells in exponential growth phase to result in completely overgrown specimens with cells in stationary growth phase. The final concentration was 10^6^ cfu mL^−1^ (vegetative and sporulated bacteria). Statistics are based on three biological and experimental replications.

### 2.3. Contamination of Specimens and Recovery of Surviving Microorganisms

#### 2.3.1. Contamination

For a first step contamination of the PE-stripes, a 4-h culture with bacteria at exponential growth stage was prepared by using 20 mL of inoculated nutrient broth. Inoculation of the broth was done by adding three colony-forming units (cfu) with an inoculation loop. The inoculated broth was not shaken during the incubation time. Therefore, bacteria in the exponential growth phase were achieved. After this first incubation step, 1 mL of the 4-h culture was used for dilution in 30 mL fresh nutrient broth. A volume of 60 mL (2 × 30 mL) was needed for the contamination of 30 PE-stripes. For each experiment, 30 PE-stripes were needed and prepared. The sterilized PE-stripes were added to 2 mL tubes with a screw cap and submerged in 1.3 mL of the diluted 4-h culture. The closed tubes were shaken 24 h at the required incubation temperature. Shaking was done with 80 rpm. Therefore, bacteria in the stationary growth phase were achieved. The inoculated PE-stripes were washed three times with PBS (phosphate buffered saline), transferred to a new 2 mL screw cap tube, and kept closed before the PTW/PPA treatment took place. The inoculation of the PE-stripes with endospores of *B. atrophaeus* was done differently, because the spores are a dormant living form and grow not actively on the stripes. For each PE-stripe 100 µL were pipetted on each side. The drying of each side required 2–3 h and was realized under laminar flow conditions. The spore-inoculated stripes were also transferred to 2 mL screw cap tubes for PTW/PPA treatment.

#### 2.3.2. Recovery of the Microorganisms

Subsequently to a fractionated or combined PTW and PPA treatment, the surviving colony-forming units were recovered. For this purpose, the treated PE strips were transferred into 10 mL nutrient broth and shaken for 15 min at 300 rpm. The resulting suspensions were used for dilutions and further investigations. By using the surface-spread-plate count method, the colony-forming units were visualized after incubation in the incubator for 12–18 h. This detection method can be used for aerobic bacteria. The volume of 100 µL of all serial dilutions of the suspension were plated out. Serial dilutions were performed as 1 to 10 dilutions. The detection limit for this method was 10 or 100 cfu mL^−1^ for the experiments described in this paper. Colony-forming units were counted manually. 

### 2.4. Plasma Processed Air (PPA) Generation

The generation of plasma-processed air (PPA) for the non-thermal treatment of the contaminated samples was carried out with a microwave driven discharge device. The arrangement used is shown in Figure 1. The used microwaves had a frequency of 2.45 GHz with an input power in the range of 1.1 kW, which results in a gas temperature of approx. 4000 K at a gas flow of 18 slm air. By cooling the PPA down to room temperature, the gas could be applied directly for decontamination. 

The main components of PPA are RNA (reactive nitrogen species) and non-processed air. RNAs are known as antimicrobial agents [14,31].

### 2.5. Chemistry of Plasma-Processed Air (PPA)

Pipa et al., (2012) performed an analysis of the microwave plasma with optical emission spectroscopy (OES) [32]. The measurements of the microwave plasma have also used air as working gas and identified ROS (reactive oxygen species) and RNS (reactive nitrogen species) within the discharge.

By comparison with known wavelength, ROS like OH radicals and RNS like NO and NO_2_ radicals were identified.

The plasma effluent was analyzed by mass spectrometry; these results were published by Schnabel et al., (2015a) [14]. The results of these analysis showed that only 2.6% of the compressed air used as working gas had been processed by the plasma. The composition of the processed gas contained mainly RNS like NO_2_ (1.8%) and NO (0.6%), smaller chemical components were HNO_2_, HNO_3_, CO_2_ and H_2_O [14,33]. Due to high discharge temperatures, Ozone (O_3_), a well-known antimicrobial effective ROS, could not be detected [16].

The specific concentration of NO_2_, which is the most detected RNS in mass spectrometry, was measured by a self-constructed NO_2_ sensor calibrated using test gas (Linde AG, Pullach, Germany: test gas NO_2_ 3.19% and synthetic air). The sensor was based on absorbance measurements of NO_2_-bands at 400 nm [34]. The plasma-on time dependent NO_2_ concentration was about 5000 ppm for 5 s on time and about 12700 ppm for 50 s [13]. However, neither NO_2_, which reacts to HNO_2_ in contact with water, nor an acidification of the treated water are exclusively responsible for the antimicrobial potential of PPA and PTW [35]. A support of the observed antimicrobial effects by other compounds hosted in PPA/PTW may be possible.

### 2.6. Plasma-Treated Water (PTW) Generation

Non-thermal plasma treatment of the contaminated specimens was done with microwave driven discharge generated PTW by contact of PPA with distilled, sterile water (Figure 2). The generated PPA was cooled down to room temperature and could react with the water (10 mL). The PTW was used in a volume of 1.3 mL for each PE-stripe. 

### 2.7. Chemical Analysis of Plasma-Treated Water (PTW)

#### 2.7.1. Ion Chromatography for Nitrite and Nitrate Detection

Immediately after preparation, the PTW was diluted with PBS buffer pH 7.4 at a ratio of 1:10 or higher. Nitrite and Nitrate were determined by ion chromatography (IC). IC was performed on a Professional IC 850 instrument (Metrohm, Switzerland). The sampler was a Sample Center 889 IC set to a constant temperature of 4 °C. The volume of the injection loop was 10 μL. A Metrosep a Supp 5–150/4.0 with guard column from the same material were used for separation. As eluent 3.2 mmol L^−1^ of Na_2_CO_3_/1 mmol L^−1^ NaHCO_3_ was used and the flow rate was 0.8 mL min^−1^. The column temperature was 20 °C. The IC was equipped with a conductivity detector and a scanning UV detector. The UV detector operates at 220 nm. 

#### 2.7.2. Photometric Detection of Nitrite

Additionally, nitrite was determined photometrical by the nitrite test Spectroquant® (Merck KGaA, Darmstadt, Germany). The spectrophotometric measurements were done at λ = 525 nm after 10 min incubation and compared to a sodium nitrite standard curve. The method is analogous to DIN EN 26777:1993-04 [36].

### 2.8. Single and Combined Treatment with PTW and PPA 

#### 2.8.1. PPA Treatment

During the individual treatment with PPA, the discharge was ignited for 5, 15 or 50 s (pre-treatment, plasma on time). The PPA was generated in these time windows. Due to the software control, the gas flux and the microwave power are limited by the pre-treatment duration. 

The amount of generated NO_2_ was measured and published before [13].

Subsequently, samples were treated 1, 3 and 5 min with the generated PPA (post-treatment time). The PPA-incubation period was stopped by evacuating the PPA in the sample-loaded reaction chamber with pure compressed air, a step which was repeated twice in a row. The treatment chamber had a volume of 4.5 L (270 × 185 × 150 mm^3^) and the samples were fixed free standing inside the chamber to keep the contact areas as small as possible. 

The detected inactivation of the microorganisms was dependent on the amount of short- and long-lived reactive chemical species produced during the pre-treatment time and the post-treatment time with PPA, itself. 

For references, the post-treatment time was indicated at 0 min, which means these samples were exposed to an airflow instead of a PPA flow.

#### 2.8.2. PTW Treatment

For PTW production, the PPA was introduced into distilled, sterile water. The resulting PTW was used solely to inactivate the contaminated PE strips. As shortly mentioned above, the discharge was ignited for 5, 15 or 50 s (pre-treatment, plasma-on time). Subsequently, the samples were treated for 1, 3 and 5 minutes with PTW (post-treatment time). A volume of 1.3 mL PTW was added to the PE strips in the screw cap tubes to start the decontamination process. To stop the post-treatment time, the PTW was decanted into a waste container and the strips were transferred to 10 mL nutrient broth with a high buffer capacity. The recovery of the surviving/proliferating microorganisms was then continued as described in Section 2.3.2. The observed inactivation of microorganisms was dependent on incubation with long-lived reactive species in the PTW and acidification of the PTW during the post-plasma treatment time.

#### 2.8.3. Combined Treatment

For the detection of synergistic effects of a combined PTW and PPA treatment, the single working steps for PTW and PPA treatment, described before in 2.8.1 and 2.8.2, were matched. 

In a first step, PTW was added to the specimen for decontamination and in a second step; these treated specimens were dried and decontaminated by PPA. Different treatment times of combined PTW and PPA were tested (Table 2).

#### 2.8.4. Calculations for the PTW and PPA Treatment

We postulate a synergistic effect that enhances the microbial effectiveness when PTW and PPA have been used in a combined manner. The following chapter focuses on that synergistically enhanced antimicrobial potential of PPA and PTW. Therefore, the experimental set up embraces measurements of the reduction factors ($\tilde{\mathrm{RF}}$) of PE-samples that have been treated separately and combined by PTW and PPA, respectively. A sample, which undergoes a combined treatment, is treated with PTW and afterwards treated by the gaseous PPA. Combined effects can be additive, synergistic or simply reveal a decreased antimicrobial efficacy (antagonistic).

The experiments were evaluated by the determination of the *RF*, which was calculated as a ratio between the concentration of the microorganisms on a reference and the concentration on a treated sample. The concentrations of microorganisms were determined on agar-plates as described above. Counted samples underwent either a separated treatment or a combined procedure. 

For treated samples, the *RF* was calculated as follows:(1)$$RF=\frac{n_{MO}^{Ref}}{n_{MO}^{Sam}}$$
$$n_{MO}^{Ref}:\;concentration\;of\;microorganismn\;of\;the\;reference$$
$$n_{MO}^{Sam}:\;concentration\;of\;microorganismn\;of\;the\;treated\;sample$$
(2)$$\tilde{RF}=\mathrm{lg}\left(RF\right)$$

The standard deviation of the reduction factor (Δ*RF*) was calculated as follows:(3)$$\Delta RF=\sqrt{{\left(\frac{\partial RF}{\partial n_{MO}^{Ref}}\cdot \Delta n_{MO}^{Ref}\right)}^{2}+{\left(\frac{\partial RF}{\partial n_{MO}^{Sam}}\cdot \Delta n_{MO}^{Sam}\left(i\right)\right)}^{2}}$$
(4)$$\frac{\partial R}{\partial n_{MO}^{Ref}}=\frac{1}{n_{MO}^{Sam}\left(i\right)}$$
(5)$$\frac{\partial R}{\partial n_{MO}^{Sam}}=-\frac{n_{MO}^{Ref}}{n_{MO}^{Sam}{\left(i\right)}^{2}}$$
(6)$$\Delta RF=\sqrt{{\left(\frac{1}{n_{MO}^{Sam}\left(i\right)}\cdot \Delta n_{MO}^{Ref}\right)}^{2}+{\left(\frac{n_{MO}^{Ref}}{n_{MO}^{Sam}{\left(i\right)}^{2}}\cdot \Delta n_{MO}^{Sam}\left(i\right)\right)}^{2}}$$
$$\Delta n_{MO}^{Ref}\;:error\;of\;reference$$
$$\Delta n_{MO}^{Sam}\left(i\right)\;:error\;of\;sample\;i$$
ΔRF :error of reduction factor

## 3. Results

### 3.1. Antimicrobial Efficacy of Single PTW/PPA Treatment and Synergistic Effects

An important factor when comparing two disinfection processes is whether they were added in the linear or logarithmic scale. This depends on the procedures themselves (here PTW and PPA treatment) and the treatment conditions. In the best case from the point of view of hygiene, both treatments act completely independent of each other. In this case, the reduction factor ($\tilde{\mathrm{RF}}$) of both treatments could be added in the logarithmic scale ($\tilde{\mathrm{RF}}$_comb_ = $\tilde{\mathrm{RF}}$_add_). In order to identify the kind of interaction, the logarithmic scale of the combined treatments ($\tilde{\mathrm{RF}}$_comb_) were compared with the sum in the logarithmic scale of the individual treatments ($\tilde{\mathrm{RF}}$_add_). Therefore, if we obtain a measured reduction factor for our combined procedure ($\tilde{\mathrm{RF}}$_comb_) which is smaller than the sum of the measured individual procedure steps ($\tilde{\mathrm{RF}}$_add_ =$\tilde{\;\mathrm{RF}}$_PTW_ +$\tilde{\;\mathrm{RF}}$_PPA_) than the individual inactivation steps do not work utterly independent from each other. This case is called antagonistic and is marked in the tables for better allocation in red ($\tilde{\mathrm{RF}}$_comb_ < $\tilde{\mathrm{RF}}$_add_). If we achieved a combined reduction factor equal to the added value ($\tilde{\mathrm{RF}}$_comb_ = $\tilde{\mathrm{RF}}$_add_) in the experiments, we have completely independent operating processes and this is called additive and indicated by yellow in the tables. If we gain additional inactivation mechanisms, which is indicated by $\tilde{\mathrm{RF}}$_comb_ > $\tilde{\mathrm{RF}}$_add_ it is called synergistic and is labelled in the tables in green. If the detection limit was reached in the microbiological experiments and the values could not be compared, this is marked in blue. The limitations of the specific application must always be taken into account when optimizing the application parameters. In most cases, this will have an influence on the applicable treatment times for the PTW and the PPA procedure or on the duration of the whole process. These acceptable treatment times define a region within the tables. Within this area, the green and blue marked values represent the optimal range, which was used for the optimal process window. Unfortunately, this is also dependent on the microorganisms, which should be inactivated by the process. As an example, four different microorganisms are displayed: *P. carotovorum* in Table 3, *L. monocytogenes* in Table 4, *E. coli* in Table 5 and endospores of *B. atropheaus* in Table 6.

#### 3.1.1. Pectobacterium Carotovorum ATCC 15713

Table 3 shows the results for the separated and combined treatments of *P. carotovorum* with PTW and PPA.

An increase of the reduction factor was observed, if both treatment times (pre- and post-treatment) were increased for single PTW and PPA application. The synergistic effect of the combined PTW/PPA treatment was pronounced by 50 s plasma-on time for PTW and 15 s as well as 50 s plasma-on time for PPA. The increase of the post-treatment time was less influential on the increase of the reduction factor.

The fact that the effect of the post-treatment time was less pronounced than that of the pre-treatment time—which was correlated mainly with the concentration of active species—showed the dominance of concentration over time. This could explain also the high efficacy of the PPA process in comparison to the PTW process. It was obviously much easier to achieve high species concentration in the gas than in the liquid phase due to the higher density of liquids.

#### 3.1.2. Listeria Monocytogenes ATCC 15313

Table 4 displays the reduction factors for *L. monocytogenes* obtained in separated and combined PTW and PPA treatments.

If both treatment times (pre- and post-treatment) were increased for single PTW and PPA application, an increase of the reduction factor could be observed. This effect was stronger for PPA treatment. The synergistic effect of the combined PTW/PPA treatment was more pronounced by 3 minutes post-treatment time of PTW as by 1 minute. For PPA a 15 s plasma-on time has shown the best synergistic results. In all combinations, no antagonistic effects could be observed.

#### 3.1.3. *Escherichia coli* K12 NCTC 10538

Table 5 gives the results for the separated and combined treatment of *E. coli* with PTW and PPA.

If both treatment times (pre- and post-treatment) were increased for single PTW and PPA application, an increase of the reduction factor could be observed. The synergistic effect of the combined PTW/PPA treatment was pronounced by 50 s plasma-on time for PTW and 15 s as well as 50 s plasma-on time for PPA. The increase of the post-treatment time also has an impact on the increase of the reduction factor. However, the blue colored fields indicated that the detection limit was reached fast during the experiments; therefore, it is difficult to classify this as an increased antimicrobial effect.

#### 3.1.4. *Bacillus Atrophaeus* Endospores ATCC 9372

Table 6 shows the reduction factors for *B. atrophaeus* endospores obtained through separated and combined PTW and PPA treatments.

If both treatment times (pre- and post-treatment) were increased for a single PTW and PPA application, an increase in the reduction factor could be observed only for the PPA treatment. The synergistic effect of the combined PTW/PPA treatment was more pronounced with shorter pre-treatment times of PPA (5s and 15s). This was contrary to the investigated vegetative bacteria. For the 50 s pre-treatment with PPA the synergistic effect changed to additive and antagonistic results.

### 3.2. Chemical Analysis of PTW

#### 3.2.1. Influence of pH Value

After dilution of the PTW with PBS buffer, which was set by pH 7.4, the pH of the mixture dropped to 6.2 or higher. Measurements of unbuffered PTW showed that the nitrite and nitrate values were not constant over the time. On the other hand, after addition of the PBS buffer, nitrite and nitrate were found in a stable equilibrium for at least 4 h.

#### 3.2.2. Nitrite Measurement

As expected, the nitrite values had a pronounced dependence on the pre-treatment time (plasma on time). Concentrations between 76.2 to 417.1 mg L^−1^ were found in photometric measurements and between 83.1 to 476.9 mg L^−1^ for an ion chromatographic analysis (Figure 3). The photometrical determined values showed a distinct correlation with values obtained from ion chromatographic measurements with regard to their error margins (Figure 4). This was unexpected, because the nitrite test Spectroquant^®^ only senses concentrations up to 2.5 mg L^−1^. Therefore, the samples have to be diluted to fit the test’s range. A procedure that introduces error sources, which are obviously negligible. In addition, the good agreement between the photometric values and the results of ion chromatography (IC) indicated solutions that do not optically disturbed the photometric nitrite determination in PTW.

#### 3.2.3. Nitrate Measurements

Consequently, the nitrate values showed also a dependency on the pre-treatment time (plasma-on time), whose mean values ranged from 210.3 to 1644.4 mg L^−1^ (Figure 5). 

In contrast to nitrite, the nitrate values determined with nitrate test Spectroquant^®^ differed significantly from IC values, which is caused by a disturbance of nitrite in the measurement solution. Therefore, Figure 5 only gives the nitrate values obtained by IC. The nitrate values were 2.7 to 3.9 times higher than the nitrite values.

## 4. Discussion

The production of food and other produce (e.g., pharmaceuticals or cosmetics) requires a high hygienic standard, which ensures a high quality and a safe usage of these products. A factor that gains importance when the produce must fulfill high microbiologic demands. Especially in the food industry, poor or inadequate hygienic surroundings lead to health infections and foodborne diseases as well as to high production losses. The high nutrient content and the relatively high pH value (pH 5.5 to 7) of some fruits and vegetables may increase the growth of pathogens [37]. Usually, bacterial and fungal contaminations of foods or their processing environments plus interrupted freezer chains are responsible for foodborne diseases, production and quality losses. Typical human pathogens found in food are *L. monocytogenes*, *Salmonella* sp., *Clostridium* sp., *E. coli*, *S. aureus*, and *Aspergillus* sp. Many molds (e.g., *Fusarium* sp.), oomycetes, *Xanthomonas* sp., *Erwinia* sp. and *Pseudomonas* sp. may parasitically live and found in any phyto-pathogen. Therefore, the inactivation of human and phyto-pathogens is the foundation for a safe consumption of any product and, thus, radiates (effects downstream) in many social and economic areas. Our chosen bacteria (*P. carotovorum*, *L. monocytogenes* and *E. coli*) and the endospores of *B. atrophaeus* cover a broad spectrum of possible food contaminations. The picked microorganisms may be responsible for human or plant diseases and for the formation of biofilms. 

In the field of conventional cleaning methods, a distinction has been made between COP (cleaning out of place) and CIP (cleaning in place). The COP process includes high-pressure cleaning in combination with foam cleaning supported by a manual cleaning of individual plant components. It is a step which is irregularly necessary. The foam used during cleaning is rinsed off with water after a defined contact time. The CIP process is often an automated cleaning process in a closed circuit, also known as recirculation cleaning. Water, steam, alkalis, acids or enzymatic solutions have been used as cleaning agents. The third possibility in the food industry is “washing in place”, which is the cleaning of different components in dipping baths or dishwashers. Furthermore, UV radiation has often been used to ensure disinfection after cleaning. Other decontamination techniques including ozone that is a potent antimicrobial agent comprise the drawback of material damage when the concentration of the agent is too high. Regular cleaning and disinfection cycles of the production plants and rooms for contaminants removal means high costs, less productivity, maybe highly polluted wastewater and material damage for the manufacturer. Hence, the food industry faces a wide range of challenges. Therefore, there is a great demand for innovative process concepts, which complements or replaces the existing systems with resource-saving, versatile and efficient technologies. Behind that background, the use of direct and indirect non-thermal plasma-based decontamination methods appears to be promising. The advantages of a decontamination agent that acts quickly, does not leave toxic residues on the food surface or in the exhaust gasses and an adjustable, product-specific process temperature compensate the disadvantage of high energy costs by the production of the PTW and PPA [38]. The promising and easy integration of antimicrobial potent species, which have been directly provided by the plasma itself or from plasma-processed compounds upstream into a non-thermal treatment mode, makes non-thermal plasmas particularly attractive for the decontamination in food processing [39]. Plasma processes are usually easy to use, inexpensive and suitable for the decontamination of products where high heat is not desired [40]. The experimental set-up described in this publication for the PTW- and PPA-based inactivation of the presented microorganisms can exclude stresses such as temperature, pressure or radiation. However, PPA and PTW embrace reactive nitrogen species such as NO and NO_2_ and HNO_2_ and HNO_3_, respectively, which may have a harmful impact on the produce’s surface. Nevertheless, only about 3% of used air has been processed and most of the water leaved untreated. If compared to their expected natural resistance and virulence, the presented antimicrobial inactivation (separate and combined) on PET-stripes will remain surprising. The different formation and composition of the cell wall and membrane of Gram-negative and Gram-positive bacteria can lead to a higher resistance of Gram-positive bacteria compared to Gram-negative bacteria to stress factors (e.g., plasma). Fungi (mold and yeasts), conidiospores and endospores can be even more resistant to such stresses [10]. However, the results present a different bacterial behavior, the Gram-positive *L. monocytogenes* has been inactivated in most cases with an additive or synergistic effect, no inhibitory case has been seen. On the other hand, the two Gram-negative bacteria *E. coli* and *P. carotovorum* showed a weaker inactivation where inhibitory cases have been also detected. A more general aspect for both types of PTW/ PPA treatment, which has been obtained for all bacteria, was the fact that an increased pre- and post-treatment time let to an increase in the decontamination rate. Another unexpected point was the inactivation kinetics of endospores of *B. atrophaeus*. Commonly, endospores of bacteria are known as the most resistant microorganisms. However, in our experiments, the combination of all PTW-treatments with short and middle PPA treatments (5 s + 3 min up to 15 s+ 5 min) was synergistic. Only the five s + 1 min treatment shows also additive effects. Extension of the PPA treatment to 50 s combined with one, three and five min lead to more additive and inhibitory effects. A previous work showed promising inactivation results for different microorganisms on polymeric surfaces and plant tissue by PPA and PTW in single use [13,35,41,42].

There are many possible mechanisms induced by PTW/PPA treatment, which may underlie the observed results. However, the effective inactivation mechanisms, and any involved signal pathways induced by direct and indirect plasma treatment are still unidentified. A first aspect of inactivation could be an acidification [43] as known for lactic acid. Causing a depression of the pH value inside a bacterial cell by ionization of the undissociated acidic molecules or a disorder of the substrate transport by an alteration of the cell membrane´s permeability may be addressed as a mechanism that cause cell inactivation. Additionally, pH values lower than 4.0 influence the growth of most foodborne bacteria negatively. Due to the chemical behavior of PTW and PPA, regardless whether applied separately or combined, an acidification on the specimen’s surface and therefore, in the direct microbial environment can be expected [14,35,44,45]. This expected acidification may lead to a similar inactivation mode as it has been observed for lactic acid. 

Another reason for the inactivation of microorganisms by PTW and PPA can be the humidity in the PPA- and PTW-process, which is provided by the ambient/compressed air or the water itself. Maeda et al., (2003) have reported about the influence of moisture in air on bacterial inactivation [46]. They observed a nearby dependency of microbiological inactivation on moisture in air for *E. coli*. The investigated microorganism showed a maximum inactivation rate at 43% relative humidity as it occurs in laboratory/room air. 

The results for *E. coli* K12 in the present work showed an increase of inactivation by increased treatment times for a separated and combined PTW and PPA treatment. However, an improvement of the detection limit would lead to more clarity about synergistic efficacy. 

The importance of the moisture of a plasma gas for the inactivation of *Bacillus subtilis* endospores and *Aspergillus niger* conidiospores was also investigated by Muranyi et al., (2008) [47]. These scientists showed reductions up to five lg within seven s at zero percent relative gas humidity and up to four lg at 80% gas humidity for the endospores. Their observation was that high amounts of gas humidity weaken the inactivation of plasma gasses. This could be an explanation for the decreasing inactivation effect, which was seen in our investigations for endospores of *B. atrophaeus*. Longer combined treatment times that ensure a longer contact time between the PTW treated spores and the PPA let to more additive and inhibitory effects.

The third and maybe most important explanatory approach may result from the chemical composition of the PTW and the PPA. Behind that background, many antimicrobial efficient reactive oxygen and nitrogen species tending into the focus. Therefore, experimental investigations of the chemical composition of PPA and PTW are crucial to understand the biochemical reactions, which are responsible for the microbial inactivation. The chemical analysis showed that reactive oxygen species (ROS) and reactive nitrogen species (RNS) were detectable in both, PTW and PPA. The detected chemical species in PPA and PTW were OH, NO, NO_2_ and NO, NO_2_, respectively. The detection of hydrogen peroxide as well as peroxynitrite in PTW will be a part of further investigations. However, it is most probably that nitrite and nitrate are only the final products of complex reaction chains in PPA and PTW. Hydroxyl radicals (OH*) are well known as strong oxidizing agents that can damage microbial cells [10,43,48]. The photo dissociation of ozone to oxygen and singlet oxygen may also be a possibility to generate hydroxyl radicals. Oxygen and singlet oxygen can react with water to hydroxyl radicals [49]. Another highly antimicrobial effective chemical, hydrogen peroxide, can be formed by the reaction of two OH* [50]. Due to the plasma set-up and used dry air (below 32% relative humidity), chemical reactions and species, which mainly based on RNS are predominantly expected in our work. Nitrogen monoxide (NO*) is generated by nitrogen and oxygen, both can be found in air. The generation of NO* further reacts to nitrogen dioxide (NO_2_*) if oxygen (O_2_) is available as reaction component. The long-living radicals NO* and NO_2_* are known for their antimicrobial effectivity. In addition, NO* may react with ozone (O_3_) to NO_2_* and O_2_. Another possible reaction of NO* and NO_2_* can form dinitrogen trioxide (N_2_O_3_), which may react with O_3_ to nitrogen trioxide radical (NO_3_*) via dinitrogen hexoxide. Peroxynitrite, assumed to be highly antimicrobial, could be a product by the reaction of NO* with a superoxide radical [43,51]. The summarized reactions are theoretical possible in dry air after a plasma ignition. Reactions that are taking place in moisture surroundings may also be taken into account. High water amounts of treated products itself or wash water residues on abiotic surfaces may provide moisture reaction chambers. NO*, O_2_ and water (H_2_O) react to nitrite (NO_2_^−^) and hydrogen (H^+^). If instead of NO*, NO_2_* reacts with O_2_ and H_2_O, H^+^ and nitrate (NO_3_^−^ are the reaction products. The reaction of N_2_O_3_, which is generated in gas and gas/water phase, with water may additionally occur and result in NO_2_^−^ and H^+^. The two radicals NO_3_* and NO_2_* form dinitrogen pentoxide in occurrence of H_2_O [43,51]. Dinitrogen pentoxide can react with water to NO_3_^−^ and H^+^. 

However, in our experimental set-up, OH radicals have not been detected. This might be due to the absence of oxygen radicals because of a restrained energy uptake (from the microwave). Another possibility could be the generation of water clusters such as (H_2_O)NO or (H_2_O)OH. 

Nevertheless, the solutions applied in the experiments showed a strong acidification, which might be a result of the long-living generated radicals NO* and NO_2_* that react with water to nitrous acid (HNO_2_) and nitric acid (HNO_3_). Usually HNO_2_ decays to H^+^ and NO_2_^−^, but a pH value beneath 2.75 could lead to a spontaneous forming of OH* and NO* radicals. Numerous of the postulated ions, radicals and molecules are known for their strong antimicrobial effects. In combination with the antimicrobial potential of a pH shift into acidic surroundings, the occurrence of such compounds, which are also influenced by the pH, may result in the observed microbial inactivation. Further investigations are necessary and will provide a better insight into the chemical and biochemical processes underlying the antimicrobial effects observed and assumed in the presented work.

An important aspect to mention here is the possible VBNC (Viable but nonculturable) status of many bacteria, which is assumed as an active bacterial response (surviving strategy) to unusual stresses. A process that is comparable to the sporulation of *Bacillus* or *Clostridium* as well as the dormancy in *Micrococcus luteus* [52,53]. This status means that bacteria do not proliferate on/in nutrient media and therefore are not detectable in common proliferation assays. However, their metabolic activity, like the intracellular ATP level or membrane potential, remains unchanged [54,55]. Contrary to this, the synthesis of macromolecules and cell breath is limited [56]. Furthermore, changes in fatty acid and protein composition of the double lipid layer (cytoplasmic membrane and cell wall) are possible [57,58,59,60]. Different triggers for VBNC status are described in literature. For instance, nutrient limitation (e.g., starvation in distilled water), salt and oxygen content, unfavorable incubation temperatures (e.g., low temperatures of 5 °C), shaking of growth containers, disinfectants and toxic metal ions (e.g., copper) are identified as VBNC inducers. Additionally, it also depends on the type of bacteria [61,62,63,64,65,66,67,68,69,70,71]. The resuscitation of VBNC cells is possible but not imperatively required [72,73]. Resuscitation, which means that the bacteria gains back the proliferative ability, is triggered by different factors divided in three categories: (1) reversal of the negative environmental factor, e.g., increasing the incubation temperature to optimum, transferring bacteria to nutrient rich media [63,64]; (2) with substances that can repair cell damage, e.g., ammonium chloride or sulphate [74,75]; (3) addition of cell-own or cell-typical signaling molecules, e.g., pyruvate, RpF protein [76,77,78,79]. During VBNC status, bacteria retain their virulence or, with regaining their cultivability, have a pathogenic effect again. In addition, antibiotic resistance has been associated with the VBNC status; especially antibiotics that are directed against the reproductive system of the bacteria have been affected [54,72,80,81,82]. Currently, literature describes for more than 60 species of bacteria the ability to change to VBNC status, including *E. coli* and *L. monocytogenes* [83,84]. Many human pathogens are included in this VBNC group, therefore, they should be recognized as a serious health risk to public health and future work should include investigations for protein synthesis (e.g., FISH) as well as Live/Dead assays (fluorescence) in combination with common proliferation assays.

However, the direct application of PTW and PPA seems to be possible, regardless of whether used separated or combined, on abiotic surfaces in contact with fresh produce. Currently, strong impacts to the fresh produce quality are not expected. First investigations with PPA on apple showed no impact on texture and appearance and a very low impact on odor. The use of PTW for lettuce washing also resulted in no negative effects on texture and color [15,85].

However, plants use radicals like ROS and RNS as a defense mechanism. This mechanism can have a more local or a wider, systemic appearance [86,87]. An important messenger in their defense signal pathways is NO* [88,89]. It has been shown by the literature that NO* is a crucial regulator of many physiological processes in plants, including stomatal closure and plant growth as well as their individual development [90,91,92,93,94,95,96]. Furthermore, NO* can regulate these processes directly by governing gene transcription. Therefore, NO*-regulated genes are included in signal transduction, defense, cell death, transport, basic metabolism and production as well as degradation of ROS. These radicals react as activators or inhibitors of enzymes, ion channels, transcription factors and therefore regulate specific processes during stress situations in plants. In the presented plasma set-up and PTW/PPA treatment, the formation of RNS, especially NO*, occurs. The observed antimicrobial effects indicate the efficiency of generated RNS. The future perspective of using PTW and/or PPA in food industry is promising, due to low-priced consumables (compressed or ambient air, drinking water, current) and high grad of adaptability for different applications (production lines). The agents can be produced on-demand, and no special training for other chemicals is needed. 

## 5. Conclusions

The aim of this work was to develop a sanitation procedure that fulfills industrial demands. Such procedures always embrace a cleaning and disinfection step. A new combined cleaning and disinfection procedure based on one source of antimicrobial agent is proposed in this work. One single plasma source, working self-supporting and on-demand can provide two different antimicrobial agents, flexible in application and combination. These combined wet and drying processes are crucial to avoid any residual humidity after the process cycle. In order to detect alterations in the chemical composition of PTW (plasma-treated water) and PPA (plasma-processed air), nitrate and nitrite measurements were performed and compared using two independent methods. By an intensive investigation of the impact of PTW and PPA separately and in combination, detailed information about the single, as well as the synergistic effects of the agents, could be obtained. The technical implementation and up-scaling of PPA/PTW-technology into industrial processes will be the future challenge.

## Figures and Tables

**Figure 1 foods-08-00055-f001:**
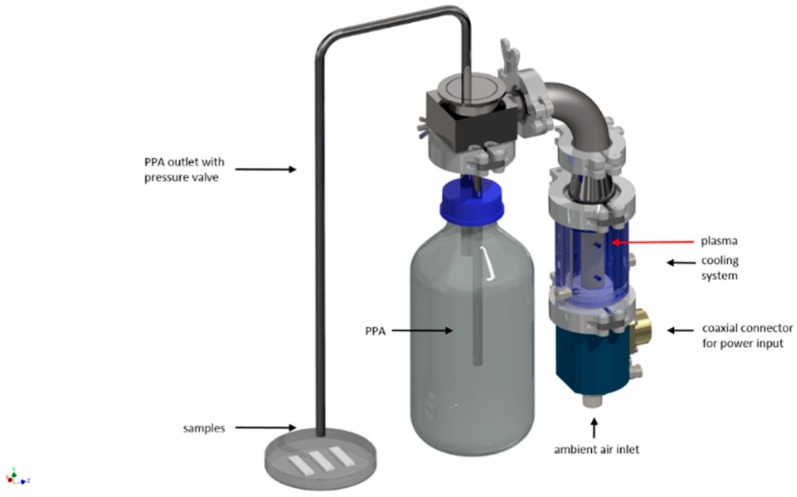
Scheme of the microwave-setup for the generation of PPA (plasma-processed air) (Baeva et al., 2014 [29], Krohmann et al., 2005 [30]).

**Figure 2 foods-08-00055-f002:**
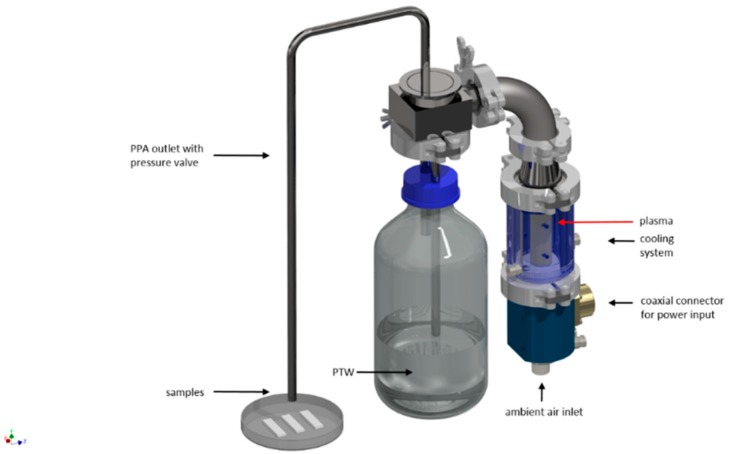
Scheme of the microwave-setup for the generation of PTW (plasma-treated water).

**Figure 3 foods-08-00055-f003:**
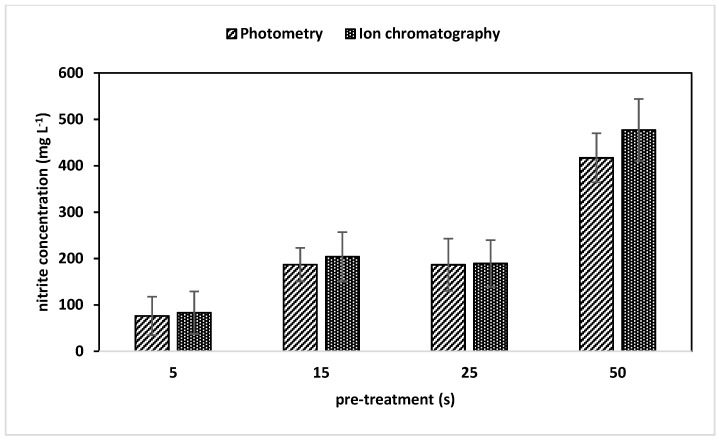
Nitrite concentrations (mg L^−1^) of PTW (plasma-treated water) with pre-treatment time (plasma on) from five up to 50 seconds measured by photometry and ion chromatography. N = 10. Standard deviations are given be error bars.

**Figure 4 foods-08-00055-f004:**
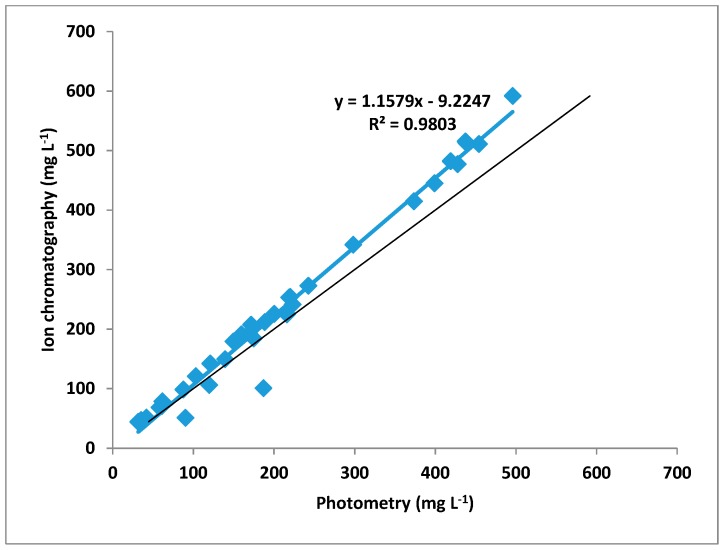
Correlation of nitrite concentration measured with photometry and with ion chromatography.

**Figure 5 foods-08-00055-f005:**
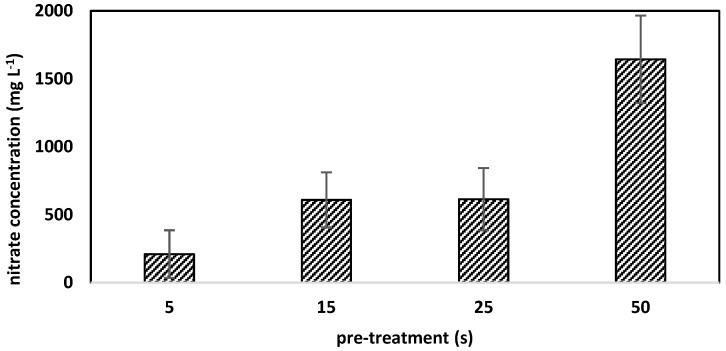
Nitrate concentrations (mg L^−1^) of PTW (plasma-treated water) with pre-treatment time (plasma on) from five up to 50 seconds measured by ion chromatography. N = 10. Standard deviations are given be error bars.

**Table 1 foods-08-00055-t001:** Overview of the used nutrient media (broth and agar) as well as incubation temperatures for the microorganisms. All media are obtained from Carl Roth GmbH + Co. KG, Karlsruhe, Germany.

Microorganism	Nutrient Broth	Nutrient Agar	Incubation Temperature (°C)
*P. carotovorum*	Standard Nutrient Broth I	Standard Nutrient Agar I	22
*E. coli* K12	TSB (trypticase soy broth)	TSA (trypticase soy agar)	37
*L. moncytogenes*	Standard Nutrient Broth I	Standard Nutrient Agar I	37
*B. atrophaeus* spores	TSB	TSA	37

**Table 2 foods-08-00055-t002:** Investigated pre- and post-treatment times of PTW (plasma-treated water) and PPA (plasma-processed air) applied separately and in combination.

PPA	PTW									
	w/o	5 s + 1 min	5 s + 3 min	5 s + 5 min	15 s + 1 min	15 s + 3 min	15 s + 5 min	50 s + 1 min	50 s + 3 min	50 s + 5 min
w/o		x	x	x	x	x	x	x	x	x
5 s + 1 min	x	x	x	s	x	x	s	x	x	s
5 s + 3 min	x	x	x	s	x	x	s	x	x	s
5 s + 5 min	x	x	x	s	x	x	s	x	x	s
15 s + 1 min	x	x	x	s	x	x	s	x	x	s
15 s + 3 min	x	x	x	s	x	x	s	x	x	s
15 s + 5 min	x	x	x	s	x	x	s	x	x	s
50 s + 1 min	x	x	x	s	x	x	s	x	x	s
50 s + 3 min	x	x	x	s	x	x	s	x	x	s
50 s + 5 min	x	x	x	s	x	x	s	x	x	s

w/o = without, single treatment with PPA or PTW; x = all investigated microorganisms (bacteria, spores); s = only endospores of *B. atrophaeus.*

**Table 3 foods-08-00055-t003:** Experimental ($\tilde{\mathrm{RF}}$_comb_) and calculated ($\tilde{\mathrm{RF}}$_add_) data of separated and combined use of PPA (plasma-processed air) and PTW (plasma treated water) under different pre- and post-treatment times for antimicrobial effects on *P. carotovorum*. Color code for effects of combined PTW/PPA treatments: red = antagonistic, yellow = additive, green = synergistic, blue = detection limit reached within the experiments. w/o = without. Exp. = experimental, Calc. = calculated.

PPA		PTW									
		w/o	5 s + 1 min	5 s + 3 min	5 s + 5 min	15 s + 1 min	15 s + 3 min	15 s + 5 min	50 s + 1 min	50 s + 3 min	50 s + 5 min
w/o	Exp.		2.46 ± 0.10	2.57 ± 0.55	2.57 ± 0.38	2.51 ± 0.18	2.56 ± 0.24	2.73 ± 0.26	2.69 ± 0.02	2.89 ± 0.26	3.01 ± 0.37
	Calc.										
5 s + 1 min	Exp.	0.87 ± 0.49	2.76 ± 0.65	2.93 ± 0.56		1.91 ± 0.53	2.82 ± 0.24		4.62 ± 0.20	3.93 ± 0.60	
	Calc.		3.33	3.44		3.38	3.43		3.56	3.76	
5 s + 3 min	Exp.	1.49 ± 0.23	4.32 ± 0.4	2.49 ± 0.42		2.85 ± 0.53	3.74 ± 0.02		4.45 ± 0.32	4.57 ± 0.00	
	Calc.		3.95	4.06		4.00	4.05		4.18	4.38	
5 s + 5 min	Exp.	1.84 ± 1.06	4.73 ± 0.00	3.73 ± 0.00		3.12 ± 0.60	4.33 ± 0.59		4.72 ± 0.00	4.57 ± 0.00	
	Calc.		4.30	4.41		4.35	4.40		4.53	4.73	
15 s + 1 min	Exp.	1.48 ± 0.16	5.00 ± 0.54	4.73 ± 0.00		3.14 ± 1.14	5.07 ± 0.69		5.42 ± 0.35	5.57 ± 0.00	
	Calc.		3.94	4.05		3.99	4.04		4.17	4.37	
15 s + 3 min	Exp.	1.47 ± 0.30	5.56 ± 0.17	4.73 ± 0.00		4.67 ± 1.08	5.96 ± 0.17		5.72 ± 0.00	5.57 ± 0.00	
	Calc.		3.93	4.04		3.98	4.03		4.16	4.36	
15 s + 5 min	Exp.	2.81 ± 0.44	5.69 ± 0.00	4.73 ± 0.00		6.39 ± 0.00	6.08 ± 0.00		5.72 ± 0.00	5.57 ± 0.00	
	Calc.		5.27	5.38		5.32	5.37		5.50	5.70	
50 s + 1 min	Exp.	1.85 ± 0.33	5.69 ± 0.00	4.73 ± 0.00		6.39 ± 0.00	5.71 ± 0.40		5.72 ± 0.00	5.57 ± 0.00	
	Calc.		4.31	4.42		4.36	4.41		4.54	4.74	
50 s + 3 min	Exp.	1.94 ± 0.22	5.69 ± 0.00	4.73 ± 0.0		6.39 ± 0.00	5.96 ± 0.17		5.72 ± 0.00	5.57 ± 0.00	
	Calc.		4.40	4.51		4.45	4.50		4.63	4.83	
50 s + 5 min	Exp.	3.64 ± 0.93	5.69 ± 0.00	4.73 ± 0.00		6.39 ± 0.00	6.08 ± 0.00		5.72 ± 0.00	5.57 ± 0.00	
	Calc.		6.10	6.21		6.15	6.20		6.33	6.53	

The following explanation should help to explain the colors in Table 3. The meaning of the colors used in Table 3 to speed up the visual acquisition of the results was the same as in Table 4, Table 5 and Table 6 below. Red color:—example for PTW with 15 s pre-treatment + 1 min post-treatment combined; with PPA with 5 s pre-treatment + 1 min post-treatment; experimental value of 2.51 ± 0.18 for single PTW treatment; experimental value of 0.87 ± 0.49 for single PPA treatment; calculated value of combined PTW/PPA treatment based on experimental; values of single treatments mean 2.51 + 0.87 = 3.38; experimental value of 1.91 ± 0.53 for combined PTW/PPA treatment; antagonistic (red) result for combined treatment as the calculated value of 3.38 is higher than the experimental value of 1.91; including standard deviation of 0.53 for experimental value of combined treatment led to a maximum of 2.44 for $\tilde{\mathrm{RF}}$, still lower than 3.38; table field is marked with red, due to antagonistic value for $\tilde{\mathrm{RF}}.$ Yellow color:—example for PTW with 5 s pre-treatment + 1 min post-treatment combined with PPA with 5 s pre-treatment + 1 min post-treatment; experimental value of 2.46 ± 0.10 for single PTW treatment; experimental value of 0.87 ± 0.49 for single PPA treatment; calculated value of combined PTW/PPA treatment based on experimental values of single treatments means 2.46 + 0.87 = 3.33; experimental value of 2.76 ± 0.65 for combined PTW/PPA treatment; antagonistic result for combined treatment as the calculated value of 3.33 is higher than the experimental value of 2.76; including standard deviation of 0.65 for experimental value of combined treatment led to a minimum of 2.11 and maximum of 3.41 for $\tilde{\mathrm{RF}}$, resemble to 3.33 as this value is in the range between 2.11 and 3.41; table field is marked with yellow, due to additive value for $\tilde{\mathrm{RF}}$. Green color:—example for PTW with 5 s pre-treatment + 1 min post-treatment combined with PPA with 15 s pre-treatment + 1 min post-treatment; experimental value of 2.46 ± 0.10 for single PTW treatment; experimental value of 1.48 ± 0.16 for single PPA treatment; calculated value of combined PTW/PPA treatment based on experimental values of single treatments means 2.46 + 1.48 = 3.94; experimental value of 5.00 ± 0.54 for combined PTW/PPA treatment; synergistic (green) result for combined treatment as the calculated value of 3.94 is lower than the experimental value of 5.00; including standard deviation of 0.54 for experimental value of combined treatment led to a minimum of 4.46 for $\tilde{\mathrm{RF}}$, still higher than 3.94; table field is marked with green, due to synergistic value for $\tilde{\mathrm{RF}}$. Blue color:—example for PTW with 5 s pre-treatment + 1 min post-treatment combined with PPA with 50 s pre-treatment + 5 min post-treatment; experimental value of 2.46 ± 0.10 for single PTW treatment; experimental value of 3.64 ± 0.93 for single PPA treatment; calculated value of combined PTW/PPA treatment based on experimental values of single treatments means 2.46 + 3.64 = 6.10; experimental value of 5.69 ± 0.00 for combined PTW/PPA treatment; detection limit (blue) for experiment was reached, no comparison between experimental and calculated $\tilde{\mathrm{RF}}$ for combined treatment possible; table field is marked with blue, due to reaching the detection limit for experimental $\tilde{\mathrm{RF}}$.

**Table 4 foods-08-00055-t004:** Experimental and calculated data of separated and combined use of PPA (plasma-processed air) and PTW (plasma-treated water) under different pre- and post-treatment times for antimicrobial effects on *L. monocytogenes*. Color code for effects of combined PTW/PPA treatments: red = antagonistic, yellow = additive, green = synergistic, blue = detection limit reached within the experiments. w/o = without. Exp. = experimental, Calc.= calculated.

PPA		PTW									
		w/o	5 s + 1 min	5 s + 3 min	5 s + 5 min	15 s + 1 min	15 s + 3 min	15 s + 5 min	50 s + 1 min	50 s + 3 min	50 s + 5 min
w/o	Exp.		0.65 ± 0.05	0.67 ± 0.02	0.71 ± 0.15	1.03 ± 0.25	0.82 ± 0.16	1.12 ± 0.34	1.35 ± 0.03	1.35 ± 0.06	1.43 ± 0.07
	Calc.										
5 s + 1 min	Exp.	0.91 ± 0.18	1.72 ± 0.31	2.60 ± 0.19		1.85 ± 0.25	2.45 ± 0.72		2.70 ± 0.29	2.63 ± 0.11	
	Calc.		1.56	1.58		1.94	1.73		2.26	2.26	
5 s + 3 min	Exp.	1.11 ± 0.08	1.93 ± 0.47	2.97 ± 0.19		1.98 ± 0.50	2.51 ± 0.59		2.85 ± 0.44	3.20 ± 0.55	
	Calc.		1.76	1.78		2.14	1.93		2.46	2.46	
5 s + 5 min	Exp.	1.24 ± 0.29	2.67 ± 0.10	3.18 ± 0.15		2.46 ± 0.44	2.58 ± 0.25		3.99 ± 0.49	4.02 ± 0.58	
	Calc.		1.89	1.91		2.27	2.06		2.59	2.59	
15 s + 1 min	Exp.	1.05 ± 0.052	2.95 ± 0.19	3.58 ± 0.15		2.97 ± 0.36	3.18 ± 0.75		5.17 ± 0.24	4.06 ± 1.01	
	Calc.		1.70	1.72		2.08	1.88		2.40	2.40	
15 s + 3 min	Exp.	1.94 ± 0.14	5.50 ± 0.28	4.06 ± 0.12		4.22 ± 0.75	3.42 ± 0.12		4.99 ± 0.39	5.20 ± 0.45	
	Calc.		2.59	2.61		2.97	2.76		3.29	3.29	
15 s + 5 min	Exp.	2.50 ± 0.08	5.72 ± 0.00	4.93 ± 0.31		5.34 ± 0.00	4.78 ± 0.00		4.90 ± 0.42	5.33 ± 0.35	
	Calc.		3.15	3.17		3.53	3.32		3.85	3.85	
50 s + 1 min	Exp.	3.25 ± 0.84	5.72 ± 0.00	5.02 ± 0.51		5.34 ± 0.00	4.78 ± 0.00		5.34 ± 0.17	5.26 ± 0.40	
	Calc.		3.90	3.92		4.28	4.08		4.60	4.60	
50 s + 3 min	Exp.	4.53 ± 0.00	5.72 ± 0.00	5.18 ± 0.49		5.34 ± 0.00	4.78 ± 0.00		5.47 ± 0.00	5.63 ± 0.00	
	Calc.		5.18	5.20		5.56	5.36		5.88	5.88	
50 s + 5 min	Exp.	4.53 ± 0.00	5.72 ± 0.00	5.35 ± 0.35		5.34 ± 0.00	4.78 ± 0.00		5.47 ± 0.00	5.63 ± 0.00	
	Calc.		5.18	5.20		5.56	5.36		5.88	5.88	

**Table 5 foods-08-00055-t005:** Experimental and calculated data of a separated and combined use of PPA (plasma-processed air) and PTW (plasma-treated water) under different pre- and post-treatment times for antimicrobial effects on *E. coli* K12. Color code for effects of combined PTW/PPA treatments: red = antagonistic, yellow = additive, green = synergistic, blue = detection limit reached within the experiments. w/o = without. Exp. = experimental, Calc. = calculated.

PPA		PTW									
		w/o	5 s + 1 min	5 s + 3 min	5 s + 5 min	15 s + 1 min	15 s + 3 min	15 s + 5 min	50 s + 1 min	50 s + 3 min	50 s + 5 min
w/o	Exp.		0.77 ± 0.11	1.01 ± 0.14	0.73 ± 0.04	1.08 ± 0.16	1.48 ± 0.27	1.41 ± 0.29	1.50 ± 0.18	1.82 ± 0.03	2.12 ± 0.11
	Calc.										
5 s + 1 min	Exp.	1.53 ± 0.13	2.03 ± 0.34	1.68 ± 0.27		1.91 ± 0.43	2.77 ± 0.37		3.80 ± 0.53	2.77 ± 1.59	
	Calc.		2.30	2.54		2.61	3.01		3.03	3.35	
5 s + 3 min	Exp.	1.44 ± 0.05	1.64 ± 0.13	1.77 ± 0.76		1.90 ± 1.37	2.79 ± 0.50		4.47 ± 0.00	3.03 ± 0.44	
	Calc.		2.21	2.45		2.52	2.92		2.94	3.26	
5 s + 5 min	Exp.	1.41 ± 0.09	3.07 ± 0.85	2.09 ± 0.96		1.96 ± 1.53	3.39 ± 0.64		4.47 ± 0.00	3.86 ± 0.96	
	Calc.		2.18	2.42		2.49	2.89		2.91	3.23	
15 s + 1 min	Exp.	2.25 ± 0.60	3.75 ± 0.49	3.00 ± 0.30		2.85 ± 1.03	4.24 ± 0.70		3.99 ± 0.39	5.37 ± 0.00	
	Calc.		3.02	3.26		3.38	3.73		3.75	4.07	
15 s + 3 min	Exp.	3.78 ± 0.60	4.23 ± 0.00	5.04 ± 0.28		4.47 ± 0.00	5.14 ± 0.00		4.47 ± 0.00	5.37 ± 0.00	
	Calc.		4.55	4.79		4.86	5.26		5.28	5.60	
15 s + 5 min	Exp.	4.42 ± 0.00	4.23 ± 0.00	5.27 ± 0.00		4.47 ± 0.00	5.14 ± 0.00		4.47 ± 0.00	5.37 ± 0.00	
	Calc.		5.19	5.43		5.50	5.90		5.92	6.24	
50 s + 1 min	Exp.	4.42 ± 0.00	4.23 ± 0.00	5.27 ± 0.00		4.47 ± 0.00	5.14 ± 0.00		4.47 ± 0.00	5.37 ± 0.00	
	Calc.		5.19	5.43		5.50	5.90		5.92	6.24	
50 s + 3 min	Exp.	4.42 ± 0.00	4.23 ± 0.00	5.27 ± 0.00		4.47 ± 0.00	5.14 ± 0.00		4.47 ± 0.00	5.37 ± 0.00	
	Calc.		5.19	5.43		5.50	5.90		5.92	6.24	
50 s + 5 min	Exp.	4.42 ± 0.00	4.23 ± 0.00	5.27 ± 0.00		4.47 ± 0.00	5.14 ± 0.00		4.47 ± 0.00	5.37 ± 0.00	
	Calc.		5.19	5.43		5.50	5.90		5.92	6.24	

**Table 6 foods-08-00055-t006:** Experimental and calculated data of separated and combined use of PPA (plasma-processed air) and PTW (plasma-treated water) under different pre- and post-treatment times for antimicrobial effects on endospores of *B. atrophaeus*. Color code for effects of combined PTW/PPA treatments: red = antagonistic, yellow = additive, green = synergistic, blue = detection limit reached within the experiments. w/o = without. Exp. = experimental, Calc.= calculated.

PPA		PTW									
		w/o	5 s + 1 min	5 s + 3 min	5 s + 5 min	15 s + 1 min	15 s + 3 min	15 s + 5 min	50 s + 1 min	50 s + 3 min	50 s + 5 min
w/o	Exp.		0.38 ± 0.12	0.31 ± 0.26	0.35 ± 0.05	0.09 ± 0.40	0.14 ± 0.04	0.31 ± 0.08	0.01 ± 0.12	0.16 ± 0.09	0.45 ± 0.17
	Calc.										
5 s + 1 min	Exp.	0.12 ± 0.03	0.73 ± 0.55	1.73 ± 0.10	1.67 ± 0.19	1.53 ± 0.18	0.03 ± 1.05	2.48 ± 0.07	2.33 ± 0.04	1.89 ± 1.00	1.28 ± 0.18
	Calc.		0.50	0.43	0.47	0.21	0.26	0.43	0.13	0.28	0.57
5 s + 3 min	Exp.	0.25 ± 0.17	1.36 ± 0.22	1.91 ± 0.17	2.07 ± 0.27	1.54 ± 0.10	1.51 ± 0.14	2.57 ± 0.17	2.31 ± 0.17	2.74 ± 0.21	1.39 ± 0.28
	Calc.		0.63	0.56	0.60	0.34	0.39	0.56	0.26	0.41	0.70
5 s + 5 min	Exp.	−0.02 ± 0.12	1.40 ± 0.37	2.07 ± 0.16	2.24 ± 0.11	1.70 ± 0.15	1.64 ± 0.29	2.70 ± 0.35	2.77 ± 0.30	2.74 ± 0.46	1.35 ± 0.09
	Calc.		0.36	0.29	0.33	0.07	0.12	0.29	−0.01	0.14	0.43
15 s + 1 min	Exp.	0.11 ± 0.06	1.11 ± 0.18	1.56 ± 0.45	1.78 ± 0.29	1.73 ± 0.1	0.80 ± 0.54	2.66 ± 0.42	2.31 ± 0.38	2.62 ± 0.10	1.13 ± 0.19
	Calc.		0.49	0.42	0.46	0.20	0.25	0.42	0.12	0.27	0.56
15 s + 3 min	Exp.	−0.11 ± 0.07	1.74 ± 0.17	1.68 ± 0.42	1.88 ± 0.21	1.65 ± 0.07	1.50 ± 0.12	2.71 ± 0.33	2.32 ± 0.24	2.93 ± 0.27	1.77 ± 0.31
	Calc.		0.27	0.20	0.24	−0.02	0.03	0.20	−0.10	0.05	0.34
15 s + 5 min	Exp.	−0.46 ± 0.50	2.23 ± 0.19	1.79 ± 0.99	2.31 ± 0.28	1.72 ± 0.15	1.93 ± 0.18	2.81 ± 0.38	2.64 ± 0.79	3.19 ± 0.26	2.03 ± 0.33
	Calc.		−0.08	−0.15	−0.11	−0.37	−0.32	−0.15	−0.45	−0.30	−0.01
50 s + 1 min	Exp.	1.04 ± 0.50	2.28 ± 0.59	1.90 ± 0.77	1.79 ± 0.09	1.7 ± 0.43	0.57 ± 0.54	2.47 ± 0.58	2.77 ± 0.71	2.71 ± 0.14	1.38 ± 0.19
	Calc.		1.42	1.35	1.39	1.13	1.18	1.35	1.05	1.20	1.49
50 s + 3 min	Exp.	1.61 ± 0.38	2.37 ± 0.54	2.07 ± 0.28	1.99 ± 0.21	1.64 ± 0.20	1.71 ± 0.15	2.52 ± 0.14	2.64 ± 0.23	2.77 ± 0.29	2.07 ± 0.51
	Calc.		1.99	1.92	1.96	1.70	1.75	1.92	1.62	1.77	2.06
50 s + 5 min	Exp.	2.74 ± 0.17	2.78 ± 0.30	2.31 ± 0.30	3.49 ± 0.51	2.13 ± 0.59	2.61 ± 0.33	3.00 ± 0.67	2.91 ± 0.63	3.57 ± 0.32	2.99 ± 0.68
	Calc.		3.12	3.05	3.09	2.83	2.88	3.05	2.75	2.90	3.19

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
