# Peer review of "Plasma-Treated Air and Water—Assessment of Synergistic Antimicrobial Effects for Sanitation of Food Processing Surfaces and Environment"

_foods, 2019, doi:10.3390/foods8020055_

Round 1
Reviewer 1 Report
Uta Schnabel, et al. investigated the synergistic antimicrobial effects of the plasma-treated air and water on the sanitation of food processing surfaces and environment. This is a well-designed experiment with a well-written manuscript and well-prepared figures. The results are solid and very persuasive. This study will facilitate the application of plasma in food processing. I only noted that the abstract of this article is too long. I recommend the authors shorten the abstract to less 200 words.
In addition, I recommend the authors change the background of Fig. 4 and Fig.5. The green background is not necessary for this paper. Please change green to white.
Author Response
Reviewer 1
Open Review
comments and Suggestions for Authors
Uta Schnabel, et al. investigated the synergistic antimicrobial effects of the plasma-treated air and water on the sanitation of food processing surfaces and environment. This is a well-designed experiment with a well-written manuscript and well-prepared figures. The results are solid and very persuasive. This study will facilitate the application of plasma in food processing. I only noted that the abstract of this article is too long. I recommend the authors shorten the abstract to less 200 words.
In addition, I recommend the authors change the background of Fig. 4 and Fig.5. The green background is not necessary for this paper. Please change green to white.
Authors are grateful to the referee for careful reading the manuscript. Revising the article in accordance with referee’s remarks allows the authors to improve the quality of the paper. This revise was done very carefully. The corresponding corrections of the text are listed in the list of changes. Author’s opinion is that the implemented changes are consistent with the referee’s intention of improvement.
The abstract has been shortened to 217 words, we hope this is acceptable.
The background has been changed to white for Figure 4 and 5.
Reviewer 2 Report
Open Review
Manuscript Number: foods-432623
Title: Plasma treated air and water – Assessment of synergistic antimicrobial effects for sanitation of food processing surfaces and environment.
General comments:
The subject of the review is interesting and very important, but there are some aspects, that could be improved and better organised.
Abstract
It has no features of Abstract. The text it is not informative; it looks like the manuscript were a review article. Authors should clearly emphasise, that this is the original article; they should assess the aim, results and conclusions of their work. The Abstract sounds like theoretical Introduction.
Detailed comments:
Introduction
Lines 52-72 and 85-105 – references should be given.
PTFE – the abbreviation should be explained, when used for the first time.
The same is connected with PPA, PPTW and others. The full names should be developed in the text for the first time, not only in the Abstract.
Materials and Methods
Table 1: the internal horizontal lines disappeared.
Figure 1 – there are 2 full stops at the end of the caption.
Table 2 – in that table and whole tables or captions, the abbreviations (PPA and PTW, and others) should be explained.
Results
Lines 299-315 – should it be rather in the methodology?
Lines 303-315 the formulas are illegible. The numbers on the right in the brackets – what they mean?
Lines 327-378 – the text is not clear; it should be justified. It does not facilitate understanding the results in the Tables.
The text should be used for description of Tables 3-6 and should be included to paragraphs 3.1.1 to 3.1.5.
Lines 379-406 – there should be the description and some commentaries of results presented in tables.
Tables 3-6 – first 2 left columns are too narrow, so the text is in the mess.
I think, that the green background in the Figures is not necessary.
Figs. 3 & 5 – the results are repeated in the tables; authors should decide how to present them – figure or table? Figures would be better. But the x axis should be better visible. In the captions the bars should be described (SD).
Conclusion
There are no clearly enunciated conclusions, which could summarise the results. What is the practical aspect of the work?
References
Within 94 references, only 10 are from/after 2015 (last 4-5 years).
Please correct reference no. 92 according to the journal demands.
Author Response
Reviewer 2
Comments and Suggestions for Authors
Title: Plasma treated air and water – Assessment of synergistic antimicrobial effects for sanitation of food processing surfaces and environment.
Authors are grateful to the referee for careful reading the manuscript. Revising the article in accordance with referee’s remarks allows the authors to improve the quality of the paper. This revise was done very carefully. The corresponding corrections of the text are listed in the list of changes. Author’s opinion is that the implemented changes are consistent with the referee’s intention of improvement
General comments:
The subject of the review is interesting and very important, but there are some aspects, that could be improved and better organised.
Abstract
It has no features of Abstract. The text it is not informative; it looks like the manuscript were a review article. Authors should clearly emphasise, that this is the original article; they should assess the aim, results and conclusions of their work. The Abstract sounds like theoretical Introduction.
The abstract has been shortened and rewritten.
Detailed comments:
Introduction
References were given wherever appropriate.
PTFE – the abbreviation should be explained, when used for the first time.
The same is connected with PPA, PPTW and others. The full names should be developed in the text for the first time, not only in the Abstract.
The abbreviations are now explained throughout the text.
Materials and Methods
Table 1: the internal horizontal lines disappeared.
This is interesting because the authors see the horizontal lines in both documents, Word and PDF. Perhaps something has been lost during editing (different table format?).
Figure 1 – there are 2 full stops at the end of the caption.
This has been changed to one full stop.
Table 2 – in that table and whole tables or captions, the abbreviations (PPA and PTW, and others) should be explained.
The abbreviations are now explained throughout the table captions.
Results
Lines 299-315 – should it be rather in the methodology?
This section has been moved in the materials and methods part, it now has the number 2.8.4.
Lines 303-315 the formulas are illegible. The numbers on the right in the brackets – what they mean?
The numbers in brackets have been deleted.
The authors used the Word formula tool to write the formulas. In both submissions (word and pdf), we see the formulas clearly. The other reviewers were also able to see them. Perhaps something was lost during the editing.
Lines 327-378 – the text is not clear; it should be justified. It does not facilitate understanding the results in the Tables. The text should be used for description of Tables 3-6 and should be included to paragraphs 3.1.1 to 3.1.5.
The authors think that this explanation of the colors on the concrete examples contributes to a better understanding of the results in the tables, as this has already been noted by reviewers as helpful.
The examples have been assigned to Table 3 (3.1.1).
Lines 379-406 – there should be the description and some commentaries of results presented in tables.
Commentaries for the tables and results have been included.
Tables 3-6 – first 2 left columns are too narrow, so the text is in the mess.
The author provided tables 3-6 with a size fitting on one page; the editing process changed the layout to two pages and smaller columns. Tables have been modified, so that the text in the columns should be clear.
I think, that the green background in the Figures is not necessary.
The authors see a white background in all versions (provided Word and PDF file as well as edited file of the journal). The authors have changed the background for Figure 4 and 5 to white and hope that this was successful.
Figs. 3 & 5 – the results are repeated in the tables; authors should decide how to present them – figure or table? Figures would be better. But the x axis should be better visible. In the captions the bars should be described (SD).
The tables have been deleted and the bars have been described within the captions.
Conclusion
There are no clearly enunciated conclusions, which could summarise the results. What is the practical aspect of the work?
The conclusion has been reworked.
References
Within 94 references, only 10 are from/after 2015 (last 4-5 years).
Please correct reference no. 92 according to the journal demands.
The reference is correctly specified in the Word and PDF manuscripts supplied. However, it is italic in the version provided by the journal for the review process. Perhaps this happened during the editing process?
Reviewer 3 Report
1.Abstract need to rewrite with clear aim, approach, results & conclusion. Authors need to rewrite the hypothesis/purpose of the work more clearly in the introduction section.
2.Need to include citation from line 52-72 (where ever it applicable).
3.Need to avoid repeated figures.
4.The presentation of the results should improve with different type of approach.
5.The manuscript need minor revisions and should be stringent manuscript, which has been carefully proof read by the authors before re-submission. However, the overall writing of the manuscript need English correction
Author Response
Reviewer 3
Comments and Suggestions for Authors
Authors are grateful to the referee for careful reading the manuscript. Revising the article in accordance with referee’s remarks allows the authors to improve the quality of the paper. This revise was done very carefully. The corresponding corrections of the text are listed in the list of changes. Author’s opinion is that the implemented changes are consistent with the referee’s intention of improvemen
1. Abstract need to rewrite with clear aim, approach, results & conclusion. Authors need to rewrite the hypothesis/purpose of the work more clearly in the introduction section.
The abstract has been shorten and rewritten.
2. Need to include citation from line 52-72 (where ever it applicable).
Citations have been included
3.Need to avoid repeated figures.
Figure 3 and 5 have been changed. Figure 1 and 2 are different.
4. The presentation of the results should improve with different type of approach.
The authors is not quite clear want is meant with this comment. However, the results section is changed due to the comments of other reviewers.
5.The manuscript need minor revisions and should be stringent manuscript, which has been carefully proof read by the authors before re-submission. However, the overall writing of the manuscript need English correction
The authors revised the English throughout the manuscript carefully and hope that the reviewer’s comments are addressed.
Round 2
Reviewer 2 Report
The Manuscript looks better now, especially the Abstract.
But there are some comments:
1. Line 135: there is‘…stainless steel, glass, polymers, rubber and PTFE (polytetrafluoroethylene))…’and it should be:‘…stainless steel, glass, polymers, rubber and PTFE – polytetrafluoroethylene)…’So as to avoid double brackets.
2. I have not checked the ‘Instructions for authors’ for the Journal, but the citations (e.g. lines 138-139; line 317; lines 948-951) are neither chronologically, nor alphabetically.
3. In Table 1 caption – authors should write in the past tense about the media (the media were obtained…).
4. Line 270: why Phosphate Buffered Saline is written in the capital letters?
5. Line 470: the subtitle ‘3.1 Antimicrobial efficacy and synergistic effects’ is not informative; i.e. antimicrobial activity of what?
6. For me Tables 3-6 are too complicated and illegible, even the explanation to tables is also complicated. The tables should be rather more commented as the text and the results demand interpretation.
7. I am not a native English speaker, but there is a mess in the tenses used by Authors. The results should be written in the simple past.
8. Overall, the manuscript demands good technical editing.
Author Response
Reviewer 2
Comments and Suggestions for Authors
The Manuscript looks better now, especially the Abstract.
But there are some comments:
1. Line 135: there is ‘…stainless steel, glass, polymers, rubber and PTFE (polytetrafluoroethylene))…’ and it should be: ‘…stainless steel, glass, polymers, rubber and PTFE – polytetrafluoroethylene)…’ So as to avoid double brackets.
This has been changed.
2. I have not checked the ‘Instructions for authors’ for the Journal, but the citations (e.g. lines 138-139; line 317; lines 948-951) are neither chronologically, nor alphabetically.
We sorted the references from latest to oldest in the manuscript and alphabetically in the reference section.
3. In Table 1 caption – authors should write in the past tense about the media (the media were obtained…).
This has been changed.
4. Line 270: why Phosphate Buffered Saline is written in the capital letters?
This has been changed.
5. Line 470: the subtitle ‘3.1 Antimicrobial efficacy and synergistic effects’ is not informative; i.e. antimicrobial activity of what?
This has been changed to “Antimicrobial efficacy of single PTW/PPA treatment and synergistic effects”.
6. For me Tables 3-6 are too complicated and illegible, even the explanation to tables is also complicated. The tables should be rather more commented as the text and the results demand interpretation.
More comments and interpretation have been included in the results section.
7. I am not a native English speaker, but there is a mess in the tenses used by Authors. The results should be written in the simple past.
The tense for the results is changed to past.
8. Overall, the manuscript demands good technical editing.